# Eco-Friendly Fabrication of Secretome-Loaded, Glutathione-Extended Waterborne Polyurethane Nanofibers

**DOI:** 10.3390/ijms262311556

**Published:** 2025-11-28

**Authors:** Paolo Accardo, Francesco Cancilla, Annalisa Martorana, Filippo Calascibetta, Giandomenico Amico, Giovanna Pitarresi, Calogero Fiorica, Cinzia Maria Chinnici, Fabio Salvatore Palumbo

**Affiliations:** 1Dipartimento di Scienze e Tecnologie Biologiche Chimiche e Farmaceutiche (STEBICEF), Università degli Studi di Palermo, Via Archirafi 32, 90123 Palermo, Italy; paolo.accardo@unipa.it (P.A.); francesco.cancilla01@unipa.it (F.C.); giovanna.pitarresi@unipa.it (G.P.); calogero.fiorica@unipa.it (C.F.); 2Cell Therapy Group, Fondazione Ri.MED c/o IRCCS ISMETT, Via E. Tricomi 5, 90127 Palermo, Italy; amartorana@fondazionerimed.com (A.M.); fcalascibetta@fondazionerimed.com (F.C.); cchinnici@fondazionerimed.com (C.M.C.); 3Regenerative Medicine and Immunotherapy Unit, Fondazione Ri.MED c/o IRCCS ISMETT, Via E. Tricomi 5, 90127 Palermo, Italy; gamico@fondazionerimed.com

**Keywords:** waterborne polyurethane, electrospinning, secretome delivery

## Abstract

The development of advanced delivery systems for bioactive factors is a critical focus in regenerative medicine and tissue engineering. In this study, we present a waterborne polyurethane (WPU)-based scaffold fabricated through a fully aqueous electrospinning process, providing a solvent-free and green method for delivering secretome derived from human mesenchymal stromal cells (MSCs). We optimized the electrospinning parameters to enable efficient secretome incorporation while preserving fiber morphology, sterility, and biocompatibility. The resulting membranes exhibited a uniform nanofibrous architecture, supported high cell viability, and demonstrated effective secretome loading and release, detected following release of vascular endothelial growth factor (VEGF)-A over 24 h. Overall, our findings highlight the potential of WPU nanofibrous scaffolds as sustainable and functional platforms for the delivery of MSC-derived bioactive factors in biomedical applications.

## 1. Introduction

The localized delivery of bioactive molecules through biomaterials represents a key strategy in tissue engineering and regenerative medicine, enabling site-specific therapeutic action that enhances efficacy while minimizing systemic exposure. Among various biomaterials, electrospun nanofibrous scaffolds have emerged as particularly promising delivery platforms owing to their high surface-to-volume ratio, tunable porosity, and structural resemblance to the extracellular matrix (ECM) [1,2,3,4]. This biomimetic architecture provides both topographical and biochemical cues that promote cell adhesion, proliferation, and lineage-specific differentiation [5,6,7,8]. Electrospun scaffolds also enable the incorporation of a wide range of therapeutic agents, including growth factors and extracellular vesicles (EVs), making them versatile tools for delivery.

Among the bioactive molecules explored, the secretome derived from mesenchymal stromal cells (MSCs) has gained significant attention as a cell-free therapeutic alternative to conventional cell transplantation. The MSC secretome comprises a mixture of trophic and immunomodulatory factors, present in both soluble form and encapsulated within EVs, that regulate key biological processes such as angiogenesis, cell migration and proliferation, immunomodulation, ECM remodeling, and inflammation [9,10,11,12]. Secretome-based therapies overcome challenges associated with cell transplantation, including potential tumorigenicity, poor engraftment, and immune rejection, while retaining potent regenerative capabilities [13,14,15,16,17]. However, direct administration of secretome is limited by rapid clearance and enzymatic degradation, highlighting the need for delivery systems that can protect, stabilize, and release bioactive factors in a controlled or sustained manner [18,19,20,21,22,23,24,25,26].

Electrospun nanofibrous scaffolds offer an effective approach to these challenges. Their high surface area and nanoscale porosity enable efficient encapsulation of secretome components, while their tunable mechanical properties, fiber diameter, and surface chemistry allow modulation of release kinetics and preservation of bioactivity [19].

Among polymers suitable for electrospinning, waterborne polyurethanes (WPUs) offer unique advantages. Unlike solvent-borne polyurethanes (PUs), which require organic solvents and emit volatile organic compounds (VOCs), WPUs are aqueous dispersible, eco-friendly, and cytocompatible [27]. Water dispersibility is achieved by incorporating polar ionic groups, such as carboxylates or tertiary amines, into the polymer backbone, allowing both synthesis and dispersion in water [27,28]. WPUs exhibit a microphase-separated structure comprising soft, flexible segments (e.g., polycaprolactone [PCL] or polyethylene glycol [PEG]) and hard, rigid segments (e.g., isophorone diisocyanate [IPDI]), conferring tunable mechanical strength, elasticity, hydrophilicity, and degradation behavior [27,28,29,30,31,32]. The introduction of polar groups into either segment allows precise control over rheological properties, aqueous dispersibility, and biocompatibility, factors that are crucial for successful electrospinning.

The synthesis of WPUs typically involves the prepolymer method, in which an isocyanate-terminated prepolymer is first formed by reacting a macromolecular polyol with a diisocyanate and an internal emulsifier such as DMPA. Before introducing the aqueous phase, the ionic groups are neutralized using a tertiary amine, and the viscosity can be fine-tuned through the minimal use of organic solvents. Upon the subsequent addition of the aqueous phase containing a chain extender, phase inversion occurs, extending the polymer chains within a heterogeneous solvent-water environment. Subsequent removal of the residual solvent yields a stable, water-dispersible polyurethane.

Depending on the incorporated polar group, WPUs can be classified as cationic, anionic, or non-ionic, and specific chain extenders (e.g., oxidized glutathione) can confer multifunctionality, biodegradability, and stimuli-responsiveness [33]. In this context, WPUs have been successfully employed to encapsulate growth factors, such as fibroblast growth factor-2 (FGF-2), through emulsion electrospinning. For instance, blends of waterborne polyurethanes and polyethylene oxide (PEO) loaded with FGF-2 produced fibrous membranes capable of sustained growth factor release, and enhanced in vitro angiogenesis, suggesting their potential as delivery systems for complex biological cocktails [34].

Redox-responsive chain extenders, such as oxidized glutathione, enable selective cleavage of disulfide bonds in reductive environments. This property is particularly relevant in tumor microenvironments, as well as in chronic inflammatory conditions such as non-healing skin wounds, where elevated glutathione levels and oxidative stress can be exploited for targeted drug delivery [27,29]. Polymers containing disulfide linkages remain stable under physiological conditions but undergo controlled degradation in pathological sites, minimizing off-target effects while allowing on-demand therapeutic release.

In our previous studies, we developed redox-responsive, degradable WPUs based on PCL-PEG-PCL triblock copolymers, DMPA, and IPDI, using oxidized glutathione as a bioinspired chain extender. These WPUs remained water-dispersible even after drying and autoclaving, exhibited redox responsiveness, and demonstrated cytocompatibility [28,35,36,37,38,39]. In the present study, we employed these synthesized WPUs to fabricate electrospun membranes. By blending WPUs with polyethylene oxide (PEO) as a carrier polymer, we enabled fully aqueous electrospinning, which enhanced fiber morphology and rheological properties. This solvent-free fabrication approach is expected to preserve the bioactivity of the encapsulated MSC secretome, thereby providing a novel and sustainable cell-free therapeutic platform. We hypothesized that integrating WPU-based electrospun scaffolds with the secretome will yield a multifunctional and environmentally friendly system for regenerative therapies [3,40,41,42,43,44,45]. To this end, we optimized the electrospinning parameters to enable the sterile fabrication of glutathione-extended WPU scaffolds that efficiently incorporate the MSC secretome. By combining the structural advantages of electrospun nanofibers with the cytocompatible and sustainable features of WPUs, this study establishes an advanced platform for the delivery of bioactive molecules, paving the way for next-generation applications in wound healing and regenerative medicine.

## 2. Results and Discussion

### 2.1. Optimization of Electrospinning Parameters for WPU Dispersions

Glutathione-extended WPUs were successfully dispersed in water through an autoclaving process, in which the solid WPU suspension was treated at 121 °C and 2 atm for 20 min. As previously reported, this treatment allows the preparation of stable aqueous dispersions with WPU concentrations of up to 30% *w*/*w* [28]. In this study, three different WPU formulations (Table 1), designated as WPU_1.0_, WPU_0.75_, and WPU_0.5_, were investigated to establish optimal conditions for aqueous electrospinning.

Initial electrospinning trials were conducted using WPU-GSSG_0.75_/PEO, blends. PEO 900 kDa was selected as the polymeric carrier due to its excellent viscosity and compatibility with WPU, which helped stabilize the electrospinning jet. PEO 900 kDa is commonly used as a carrier polymer to enhance the spinnability of water-soluble polysaccharides. In our system, the aqueous dispersions of WPU-GSSG_0.75_ did not provide sufficient chain entanglement to enable stable electrospinning, as confirmed by preliminary experiments. The addition of a small amounts of high-molecular-weight PEO, present at a concentration ten times lower than that of WPU-GSSG, provides the required entanglements and allows successful and continuous electrospinning. An O-ring electrode was also connected via a clamp to the same high-voltage power supply as the needle. Carrying the same positive charge as the jet, the O-ring effectively directed the jet toward the rotating collector, thereby enhancing its stability and increasing the length of the straight jet segment [1,3,41].

Figure 1A illustrates the WPU dispersion procedure prior to electrospinning, while Table 2 summarizes the electrospinning parameters used during the optimization. The initial experiments aimed at optimizing the WPU-to-PEO 900 kDa ratio and the total polymer weight percentage (*w*/*w*) in the dispersion to achieve a viscosity required to balance electrostatic and cohesive molecular forces, ensuring jet stability and adequate solvent evaporation during fiber formation. Once these dispersion conditions were established, electrospinning parameters were refined in subsequent experiments.

Process parameters, including flow rate (Q), applied voltage (V), and spinneret-to-collector distance (d), were systematically adjusted to maintain a stable Taylor cone, optimize jet stability, and prevent whipping instabilities. Additionally, these parameters were tuned to enhance fiber elongation, while ensuring sufficient solvent evaporation for proper solidification. The formation of uniform, consistent fibers was achieved by carefully balancing these competing forces, resulting in a scaffold with well-defined and reproducible characteristics. SEM was used to monitor the electrospinning process and assess the fiber morphology of each scaffold, guiding the optimization of process parameters and the determination of the ideal WPU-GSSG_0.75_/PEO ratio and dispersion weight (*w*/*w*). As shown in Figure 1B, which includes SEM images of representative batches, the optimization process was gradual. In batches 1–3, low WPU and carrier concentrations resulted in low-viscosity dispersions, which were insufficient to establish polymer chain entanglements between WPU-GSSG_0.75_ and PEO, thereby destabilizing the electrospinning process. This instability was evidenced by particle aggregates, tiny fibrils, and the absence of well-formed fibers (batch 3, Figure 1B). Increasing the WPU-GSSG_0.75_ concentration resulted in improved spinnability, as observed in batch 4, where elongated fibers were present, although numerous beads and microscale spots indicated residual droplet deposition. By maintaining a nearly constant WPU-GSSG_0.75_ concentration and increasing the carrier concentration from 0.7% to 1.4% *w*/*w*, electrospinning stability improved significantly. In batch 7, well-formed, continuous fibers were obtained, though residual bead formation suggested that the process was not yet fully optimized. The most stable electrospinning parameters were achieved in batch 12, where increasing the carrier concentration to 22.5% *w*/*w* yielded uniform, bead-free fibers. Further attempts (batches 16 and 17) explored whether reducing the applied voltage while maintaining a WPU-GSSG_0.75_ concentration of 22.5% *w*/*w* and a carrier concentration of 2.25% *w*/*w* would be feasible. However, lowering the voltage required a substantial reduction in flow rate, which negatively impacted fiber formation. Dimensional distribution analysis of scaffolds from batches 12, 13, and 14 (Figure 1C) confirmed that fiber sizes were in the nanometer range. Specifically, WPU-GSSG_0.75_-based electrospun fibers exhibited an average diameter of 514.3 ± 12.4 nm, with batch 14 displaying the lowest standard deviation, indicating the highest fiber uniformity. Overall, increasing the WPU-GSSG_0.75_ concentration up to 22.5% *w*/*w* improved the efficiency of the electrospinning process and the quality of the resulting fibers. Once the WPU and carrier concentrations were optimized, further enhancement in fiber quality was achieved by adjusting additional electrospinning parameters, such as applied voltage, feeding volume and spinneret-collector distance.

Building on these results, the optimization of electrospinning parameters for WPU-GSSG_0.5_ and WPU-GSSG_1.0_ was further investigated (Table 3 and Table 4, and Figure 2). Table 3 summarizes the production trials aimed at standardizing electrospinning parameters for WPU-GSSG_0.5_/PEO blends. Following multiple preliminary tests with varying polymer concentrations, the optimal formulation was identified as 22.5% WPU-GSSG_0.5_
*w*/*w* and 2.25% PEO *w*/*w*. Subsequent experiments focused on refining applied voltage, flow rate, and spinneret-to-collector distance to ensure a stable and reproducible electrospinning process. SEM analyses confirmed that applying the same dispersion parameters used for WPU-GSSG_0.75_ /PEO also yielded high-quality electrospun scaffolds (Figure 2A).

Dimensional distribution analysis of WPU-GSSG_0.5_/PEO electrospun fibers revealed an average diameter of 483.2 ± 47.9 nm (Figure 2B). In contrast, WPU-GSSG_1.0_/PEO dispersions exhibited substantial different behavior compared to WPU-GSSG_0.75_/PEO and WPU-GSSG_0.5_/PEO. At concentrations similar to those used for WPU-GSSG_0.5_ and WPU-GSSG_0.75_, the dispersions lacked homogeneity and were unsuitable for electrospinning. Even at concentrations below 17% *w*/*w*, stable conditions could not be achieved. Table 4 summarizes representative processing parameters, while Figure 2C shows the corresponding SEM image of the resulting electrospun material. This behavior can be attributed to the higher ratio of the diol PCL-PEG-PCL (Table 1), which reduces the water affinity of WPU-GSSG_1.0_, leading to high-viscosity dispersions even at low concentrations. The increased viscosity disrupted the droplet elongation phase, preventing the formation of a stable Taylor cone. The electrospun membranes (eWPU-GSSG_0.75_/PEO and eWPU-GSSG_0.5_/PEO), produced under the optimized conditions shown in Table 5, exhibited a whitish appearance and an elastic texture (Figure 3A).

Under the optimized conditions, both eWPU-GSSG_0.75_/PEO and eWPU-GSSG_0.5_/PEO membranes exhibited comparable fiber dimensions, averaging approximately 500 nm (Figure 3B). Notably, eWPU-GSSG_0.75_/PEO fibers (514.3 ± 12.4 nm) were slightly larger than those obtained from eWPU-GSSG_0.5_/PEO (483.2 ± 47.9 nm), which is consistent with the higher viscosity of the WPU-GSSG_0.75_/PEO spinning dispersion compared to that of WPU-GSSG_0.5_/PEO. As illustrated in Figure 3C, the swelling degree of the eWPU-GSSG_0.5_/PEO membrane immersed in DPBS initially increased during the first two time points, reaching its peak at 1 and 7 days, before progressively decreasing at 14 and 30 days. Figure 3C also shows the weight recovery profile (%). After 1 day of incubation, approximately 60% of the scaffold’s initial weight was recovered, indicating partial material dissolution. Between 7 and 14 days, the weight recovery further decreased to 50% and then remained stable thereafter. Based on these results, the significant reduction in swelling observed after 14 days can be attributed to polymer mass loss, which in turn reduces the swelling capacity. In contrast, eWPU-GSSG_0.75_/PEO exhibited a lower swelling degree after 1 day compared to eWPU-GSSG_0.5_/PEO, likely due to its higher hydrophobicity. Over the 30-day incubation period, a gradual increase in swelling was observed, with values rising from 300% to 500%.

Consistent with its enhanced hydrophobic nature, eWPU-GSSG_0.75_/PEO exhibited significantly lower weight loss compared to eWPU-GSSG_0.5_/PEO, retaining approximately 70% of its initial mass after 30 days. This finding highlights the superior hydrolytic resistance of eWPU-GSSG_0.75_/PEO compared to the eWPU-GSSG_0.5_/PEO membrane, which is attributable to its higher PCL-PEG-PCL segment content. Accordingly, the residual weight percentage after 30 days ranged from 70 to 75% for eWPU-GSSG_0.75_/PEO and 50–60% for eWPU-GSSG_0.5_/PEO. Additionally, Figure 3C demonstrates that the presence of DTT did not significantly accelerate hydrolysis of electrospun WPU-based scaffolds compared to those incubated in DPBS alone. Although DTT reduces disulfide bonds in glutathione (converting GSSG to GSH), the strong interactions among the hydrophobic polyurethane domains prevent rapid matrix degradation. This behavior aligns with previous findings [28], where WPU-GSSG_0.5_ and WPU-GSSG_0.75_ membranes, despite showing redox-responsive drug delivery properties, exhibited similar structural stability in both reducing and non-reducing conditions. The hydrophobic coil structures formed by the soft polyurethane segments contribute to the maintenance of biomaterial stability, regardless of the redox state of the glutathione, while still allowing effective chemical reduction of GSSG in the presence of a reducing agent.

### 2.2. In Vitro Cytocompatibility Test

The eWPU-GSSG_0.5_/PEO membranes were selected for further experiments because their faster hydrolytic degradation, compared to eWPU-GSSG_0.75_/PEO membranes, may facilitate quicker reabsorption during repeated wound-dressing changes. The cytocompatibility results for eWPU-GSSG_0.5_/PEOs are presented in Figure 4.

The MTS assay indicated cell viability of 87% at 24 h and 96% at 72 h. Live/dead staining with FITC and acridine orange/ethidium bromide after indirect contact with the membranes confirmed these findings (Figure 4B). Together, these results demonstrated the excellent cytocompatibility of the eWPU-GSSG_0.5_/PEO membranes.

### 2.3. Secretome Loading and in Vitro Release

To assess whether the incorporation of secretome into the WPU-GSSG_0.5_/PEO dispersion affected its viscoelastic properties, and consequently the electrospinning parameters and fiber formation, the rheological properties of the unloaded dispersion were compared with those of the same formulation loaded with secretome at 20-fold and 80-fold concentrations. The zero-shear-rate viscosity of the WPU-GSSG_0.5_/PEO dispersion (Figure 5A) showed no significant differences among the samples, suggesting that secretome loading did not alter intermolecular entanglement within the system. The electrospinning procedure was conducted under a laminar flow hood, using a customized enclosure to minimize air turbulence and ensure process stability (Figure 5B).

Due to variations in environmental conditions, optimal electrospinning was achieved by adjusting the flow rate to 0.01 mL/min and setting the voltage and spinneret-to-collector distance to 24 kV and 20 cm, respectively (Table 6). SEM images and fiber diameter analysis (Figure 5C,D) revealed that increasing secretome concentration influenced electrospinning efficiency, resulting in fibers with larger diameters. Membranes loaded with a 20-fold concentrated secretome exhibited small and uniform fibers, whereas those loaded with a 40-fold concentration showed greater heterogeneity, consisting of both small (thin) and large (thick) fibers. At the highest secretome concentration (80-fold), fibers became larger but regained uniformity. The stability of zero-shear viscosity suggests that the cohesive forces within the polymer dispersion are not substantially altered. However, successful electrospinning depends not only on viscosity but also on the balance between cohesive forces and the electrostatic forces acting on the jet. Cytokines and growth factors contain both acidic and basic amino acids, and their incorporation into the spinning solution may influence its overall charge distribution. Under our processing parameters, it is possible that the introduction of these charged biomolecules disrupted the charge balance sufficiently to prevent the establishment of a new stable equilibrium that would support efficient jet elongation. The observed increase in fiber diameter may therefore reflect a reduction in the net charge density of the electrospinning solution, likely due to the polyelectrolytic nature of the polypeptide components in the secretome. A lower charge density would reduce the stretching forces acting on the jet, resulting in thicker fibers even in the absence of viscosity changes. At a 20-fold concentration, these effects appear minimal, resulting in a fiber diameter distribution comparable to that obtained in the absence of secretome under standard electrospinning conditions.

VEGF-A was used as a representative growth factor of secretome content to test the release profile from WPU-GSSG_0.5_/PEO@20 electrospun membranes. Additionally, fiber diameter analysis indicated that scaffolds prepared with the 40-fold secretome concentration exhibited greater heterogeneity, whereas loading the 80-fold concentration produced excessively large fibers. Quantification of the released secretome revealed a VEGF-A loading efficiency of 93% relative to the initial amount incorporated into the electrospinning dispersion. The results indicated that the entire amount of VEGF-A loaded was released within 1500 min (Figure 5E). VEGF-A was used as a representative factor to validate both the electrospinning process and the loading capability of the material. The release profile observed in this work is faster than what we previously reported for an HA-based hydrogel, in which a plateau of VEGF release in non-heparinized hydrogels was reached after approximately 150 h (9000 min) [21]. Compared with the hydrogel system, the faster release observed here is influenced by both the amount and the modality of loading (impregnation versus incorporation during the fabrication process). It is also strongly affected by the high surface-to-volume ratio typical of electrospun fibrous scaffolds, which shortens the diffusional path length relative to bulk 3D hydrogels.

However, a more sustained and controllable release would be advantageous for many tissue-regeneration applications. Achieving such modulation will likely require the design of composite electrospun scaffolds incorporating glycosaminoglycans (e.g., heparin) or other affinity-based components capable of binding and retaining cytokines.

## 3. Materials and Methods

### 3.1. Materials

Poly(ethylene glycol) (PEG MW 1000), poly(ethylene oxide) (PEO 900 kDa), ε-caprolactone (-CL), oxidized L-glutathione (GSSG), DL-dithiothreitol (DTT), lithium bromide (LiBr), dimethyl sulfoxide (DMSO), toluene, diethyl ether, isopropanol, calcium hydride (CaH_2_), stannous octoate, sodium chloride (NaCl), sodium azide (NaN_3_), triethylamine (TEA), toluidine blue O (TBO), glycine, and Dulbecco’s phosphate-buffered saline (DPBS) were purchased from Sigma-Aldrich S.r.l. (Milan, Italy). Methyl ethyl ketone (MEK), 2,2-bis(hydroxymethyl)propionic acid (DMPA), isophorone diisocyanate (IPDI), and acetic acid were obtained from Fisher Scientific Italia (Segrate, Italy). Dulbecco’s Modified Eagle Medium (DMEM), fetal bovine serum (FBS), trypsin, L-glutamine, penicillin, streptomycin, fluorescein isothiocyanate (FITC), acridine orange (AO), ethidium bromide (EB), human dermal fibroblasts and sterile culture plates were purchased from Euroclone Group (Pero, Italy). The CellTiter 96# One Cell Proliferation Water Solution Assay (MTS) was purchased from Promega (Madison, WI, USA). Amicon Ultra-15 Centrifugal filters with a 3 kDa cut-off were purchased from Millipore (Billerica, MA, USA).

### 3.2. Apparatus

^1^H NMR spectra were recorded using a Bruker Avance II 400 MHz spectrometer (Bruker Corporation, Billerica, MA, USA) with d_6_-DMSO for WPUs and CDCl_3_ for PCL-PEG-PCL. SEC analyses were performed using an Agilent 1260 Infinity multidetector GPC/SEC system with a linear Phenogel column from Phenomenex (Milan, Italy) (5 μm particle size, 100 nm and 1 μm pore sizes). The calibration curve was determined using polystyrene standards (range 1.8–500 kDa). Measurements were performed at 50 °C ± 0.1 °C using DMF/LiBr 0.1% as eluent at a flow rate of 0.8 mL/min. UV spectra were recorded with a Shimadzu UV 2401PC spectrometer (Shimadzu, Duisburg, Germany). Nanofibers were fabricated using a custom-built electrospinning instrument consisting of a syringe pump, a high-voltage generator, a stainless-steel collector plate, and an electrospinning hood. Cell viability by the MTS assay was evaluated using a PlateReader from Eppendorf (Milan, Italy). VEGF-A was quantified using Luminex xMAP technology (Luminex 200; Luminex Corp., Austin, TX, USA) [21]. Rheological analyses were performed using a DHR-2 TA Instruments oscillatory rheometer, equipped with a parallel-plate geometry (8 mm diameter, radial grooves) and a self-heating Peltier plate (TA Instruments—Waters, New Castle, DE, USA).

### 3.3. Synthesis of GSSG-Extended WPU Urea Derivatives

A PCL-PEG-PCL triblock copolymer was synthesized following a previously reported procedure [45]. GSSG-extended WPU-urea derivatives were synthesized following a previously reported procedure [28]. Briefly, three different diol/emulsifier (PCL-PEG-PCL/DMPA) mixtures with varying molar ratios (1:1, 0.75:1.25, and 0.5:1.5) were prepared, while maintaining the molar ratios of IPDI and GSSG constant at 3:1.5, respectively. Specifically, PCL-PEG-PCL and DMPA were stirred at 70 °C and 180 rpm to ensure complete melting. IPDI was then added, and the reaction was allowed to proceed for 3 h at 70 °C under an argon atmosphere, using stannous octoate as the catalyst. After this step, a small amount of MEK was introduced, and the carboxyl groups were neutralized with TEA. Subsequently, a neutralized aqueous GSSG solution was added to initiate emulsification under vigorous stirring (1500 rpm) for 20 h. Residual MEK and TEA were removed via rotary evaporation. The product was purified by dialysis (MW cut-off 3500 Da) in ultrapure water containing NaCl for 2 days, followed by dialysis in ultrapure water for an additional 4 days. The resulting aqueous WPU dispersion was freeze-dried and stored for further use. A stable and homogeneous colloidal dispersion was obtained by resuspending the dried WPU in water at concentrations ranging from 10 to 300 mg/mL, followed by autoclaving at 120 °C for 20 min.

### 3.4. Fabrication of WPU-Based Nanofibers by Electrospinning and Characterization

To establish optimal electrospinning conditions, aqueous dispersions of WPU and PEO 900 kDa, used as a polymeric carrier, were prepared at concentrations ranging from 12.8% (*w*/*w*) to 24.8% (*w*/*w*), with PEO concentration ranging between 0.58% (*w*/*w*) and 2.26% (*w*/*w*). These ratios were consistently applied for all three WPU dispersions (labeled as 1.0, 0.75, and 0.5) corresponding to the molar ratio of PCL to the total amount of diols (PCL plus DMPA) used in the polymerization step. The resulting spinning solutions were indicated as WPU-GSSG_1.0_/PEO, WPU-GSSG_0.75_/PEO and WPU-GSSG_0.5_/PEO, respectively. Specifically, the proper amount of WPU was mixed with PEO and ultrapure water. The mixture was homogenized by vortex agitation for a few minutes, followed by autoclaving at 121 °C for 20 min. After autoclaving, ultrapure water was added to compensate for evaporation losses. The dispersion was then vortexed again and allowed to equilibrate for 24 h under orbital shaking at 37 °C to ensure complete solubilization and fluidization of the components. The spinning solutions were loaded into a 5 mL syringe equipped with an 18G blunt-tip needle and electrospun under the following conditions: flow rate of 0.008–0.023 mL/min, applied voltage of 20–30 kV, and a tip-to-collector distance of 15–25 cm. Upon completion, the electrospun samples were collected and labeled as eWPU-GSSG_1.0_/PEO, eWPU-GSSG_0.75_/PEO and eWPU-GSSG_0.5_/PEO, respectively. The morphology of the resulting nanofibers was investigated using scanning electron microscopy (SEM), and the acquired micrographs were analyzed with the ImageJ (version v1.53e) to determine fiber diameter distribution and evaluate structural uniformity. Swelling and hydrolytic degradation studies were conducted on circular specimens (10 mm diameter) of eWPU-GSSG_0.75_/PEO and eWPU-GSSG_0.5_/PEO membranes, obtained from the electrospun mats using a punch cutter. For the swelling studies, the samples were dried and weighed to determine their dry weight (w_0_). Each sample was placed into a 48-well sterile culture plate and immersed in 1 mL of DPBS or in DPBS supplemented with 10 mM DTT. The plate was incubated at 37 °C, under orbital shaking. Measurements were performed in triplicate at five time points: 1 h, 1 day, 7 days, 14 days, and 30 days. At each point, excess surface water was gently removed from each scaffold using filter paper, and the samples were reweighed to obtain the wet weight at time t (w_t_). The swelling ratio was calculated as the percentage increase in weight relative to the initial dry weight (w_0_) using the following equation:(1)$$\%\;swe=\frac{w_{t}-w_{0}}{w_{0}}\times 100$$

For hydrolytic degradation studies, scaffolds were first weighed to determine the initial weight (w_0_), placed into a 48-well sterile culture plate, and immersed in 1 mL of DPBS or DPBS supplemented with 10 mM DTT. The plate was incubated at 37 °C under orbital shaking. Measurements were conducted in triplicate at five time points: 1 h, 1 day, 7 days, 14 days, and 30 days. At each interval, the samples were washed with bidistilled water, lyophilized again, and weighed to obtain the dry weight at time t (w_t_). The degradation rate was calculated as the percentage ratio of the dry weight at time t (w_t_) to the initial dry weight (w_0_) using the formula:(2)$$\%\;deg=\frac{w_{t}}{w_{0}}\times 100$$

### 3.5. In Vitro Cytocompatibility Tests

Cytocompatibility of electrospun membranes was evaluated using human dermal fibroblasts (Euroclone Group, Milan, Italy) cultured in DMEM supplemented with 10% FBS, 2 mM L-glutamine, 100 U/mL penicillin, and 100 µg/mL streptomycin. The cells were maintained at 37 °C in a humidified atmosphere containing 5% CO_2_. eWPU-GSSG_0.75_/PEO and eWPU-GSSG_0.5_/PEO scaffolds, manufactured under sterile conditions, were rinsed in DPBS and transferred into cell culture inserts equipped with a porous membrane. These inserts were placed in 24-well sterile culture plates, and cells were seeded at a density of 1.5 × 10^4^ cells/well. Cell viability was assessed at 24 and 72 h using the MTS colorimetric assay, following the manufacturer’s instructions. At each time point, the culture medium was removed, and the wells were rinsed with sterile DPBS. An MTS solution (20% *v*/*v* in DMEM) was then added, and the plates were incubated at 37 °C for 1 h. Absorbance at 492 nm was measured using a microplate reader. Cell viability was expressed as the percentage ratio of absorbance at 492 nm in cells exposed to sterile electrospun WPU membranes relative to the absorbance of the control group, consisting of cells incubated with medium alone. To validate the results, the cell–matrix interactions were examined using fluorescein isothiocyanate (FITC), and viability was further verified through acridine orange (AO)/ethidium bromide (EB) staining.

### 3.6. Fabrication of Secretome-Loaded Electrospun Membranes

Human fetal dermal MSCs (FD-MSCs) were isolated from fetal skin according to a protocol approved by the IRCCS ISMETT Institutional Research Review Board (IRRB/00/15) and the ethics committee, using a non-enzymatic tissue outgrowth technique. Cells were cultured in DMEM supplemented with 10% FBS until they reached 80% confluence. Then, the medium was replaced with serum-free alpha-MEM, and the secretome was collected 48 h later. The collected secretome was then centrifuged at 2000× *g* for 10 min to remove cellular debris. Aliquots of secretome from independent donors were pooled and concentrated 80-fold using Amicon Ultra-15 centrifugal filters with a 3 kDa cut-off, as previously described [21]. Forty-fold and 20-fold concentrated secretome were prepared by diluting the 80-fold concentrated secretome with alpha-MEM. VEGF-A protein content was quantified using Luminex xMAP technology [21].

To prepare secretome-loaded spinning dispersions, 600 mg of WPU-GSSG_0.5_ was mixed with 60 mg of PEO 900 kDa and dispersed in water by autoclaving reaching a concentration of 22.5% and 2.26%) [28]. Then, 150 μL of secretome dispersion (20-, 40- or 80-fold concentrated) was added under a laminar flow hood to obtain WPU-GSSG_0.5_/PEO@20, WPU-GSSG_0.5_/PEO@40, WPU-GSSG_0.5_/PEO@80 dispersions at a final concentration of WPU equal to 22.5% (*w*/*w*) and PEO equal to 2% (*w*/*w*). Prior to electrospinning, the rheological properties of WPU-GSSG_0.5_/PEO, WPU_0.5_/PEO@20× and WPU-GSSG_0.5_/PEO @80× (all containing 22.5% *w*/*w* WPU 0.5 and 2% *w*/*w* PEO) were evaluated. Specifically, steady and oscillatory shear measurements were performed using a cone-plate geometry (CP50-1) with a 50 mm diameter and a 1.009° contact angle. The temperature was maintained at 22 °C using a Peltier plate. To prevent water evaporation and maintain a saturated atmosphere around the cone-plate geometry, a cover cup was placed over the system. Viscosity measurements were conducted under steady shear flow at a shear rate ranging from 10^−3^ s^−1^ to 10^3^ s^−1^, and values were considered reliable when the torque exceeded 1 μN·m. The experimental data were fitted to the Carreau–Yasuda model to calculate the zero-shear viscosity (η_0_). To ensure a controlled suspension structure, all oscillatory measurements were preceded by a pre-shear at a steady shear rate of 10^2^ s^−1^, followed by a 1 min rest phase. Oscillatory tests were then performed over a frequency range of 0.01 rad·s^−1^ to 100 rad·s^−1^. The WPU-GSSG_0.5_/PEO@20, WPU-GSSG_0.5_/PEO@40, WPU-GSSG_0.5_/PEO@80 dispersions (WPU 22.5% *w*/*w* and PEO 2% *w*/*w*) were electrospun under a laminar flow hood using a custom-made Plexiglas enclosure to minimize air turbulence and maintain stable electrospinning conditions. eWPU-GSSG_0.5_/PEO@20, eWPU-GSSG_0.5_/PEO@40, eWPU-GSSG_0.5_/PEO@80 membranes were characterized by SEM and fiber diameter distributions were quantified by ImageJ analysis.

### 3.7. Secretome Release Study

To assess the release of secretome from eWPU-GSSG_0.5_/PEO@20 membranes, disks with a 10 mm diameter were soaked in 400 µL of DPBS. At predetermined time points (60, 360, and 1440 min), aliquots of 400 µL of secretome-containing medium were collected and immediately replaced with an equal volume of DPBS. The amount of VEGF-A in the released medium was quantified using a Luminex assay. The cumulative release was calculated by summing the amounts of VEGF-A released at each time point relative to the initially loaded amount.

### 3.8. Statistical Analysis

All results are reported as mean ± standard deviation and, when applicable, statistical significance was evaluated using Student’s *t*-test (Microsoft Excel *t*-test function), assuming two-sample unequal variance and a two-tailed distribution. *p*-values < 0.05 were considered statistically significant.

## 4. Conclusions

A green, aqueous-based electrospinning process was successfully developed to produce waterborne polyurethane membranes capable of incorporating MSC secretome. Among the tested blends, WPU-GSSG_0.5_/PEO dispersions were identified as the optimal blend due to the higher hydrophilicity of WPU-GSSG_0.5_. The resulting electrospun membranes exhibited a uniform nanofibrous morphology and controlled swelling behavior. Secretome loading was efficient, with high VEGF-A loading, although release occurred rapidly within 24 h. These preliminary results demonstrate the potential of redox-responsive WPU membranes as smart delivery platforms for bioactive factors. Ongoing work is focused on optimizing scaffold composition and electrospinning parameters to achieve a more sustained release profile, with the goal of developing functional biomaterials for wound healing applications.

## Figures and Tables

**Figure 1 ijms-26-11556-f001:**
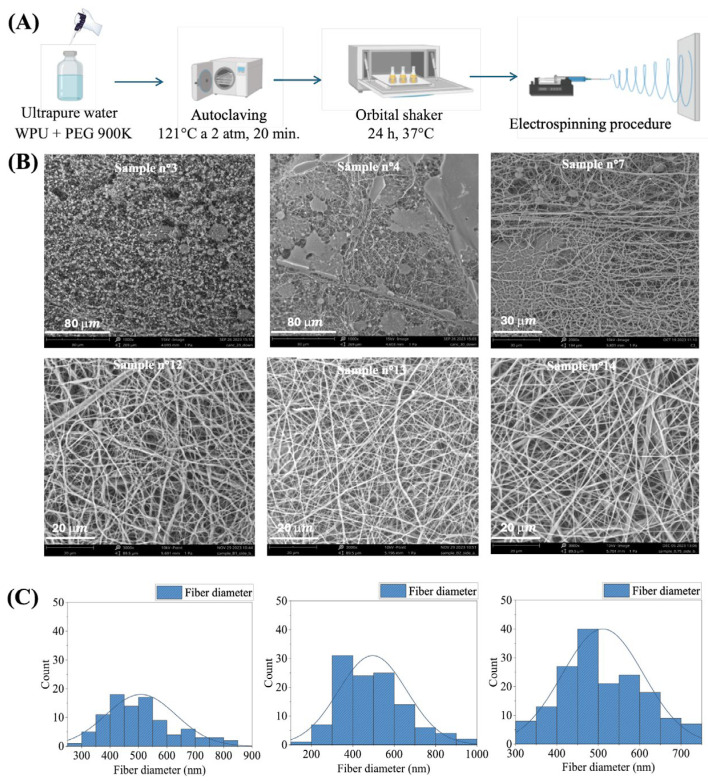
(**A**) Schematic representation of the WPU dispersion procedure. (**B**) SEM images of representative electrospun samples. (**C**) Fiber diameter distribution of the electrospun membranes calculated from SEM images using ImageJ (version 1.53e) (*n* = 200).

**Figure 2 ijms-26-11556-f002:**
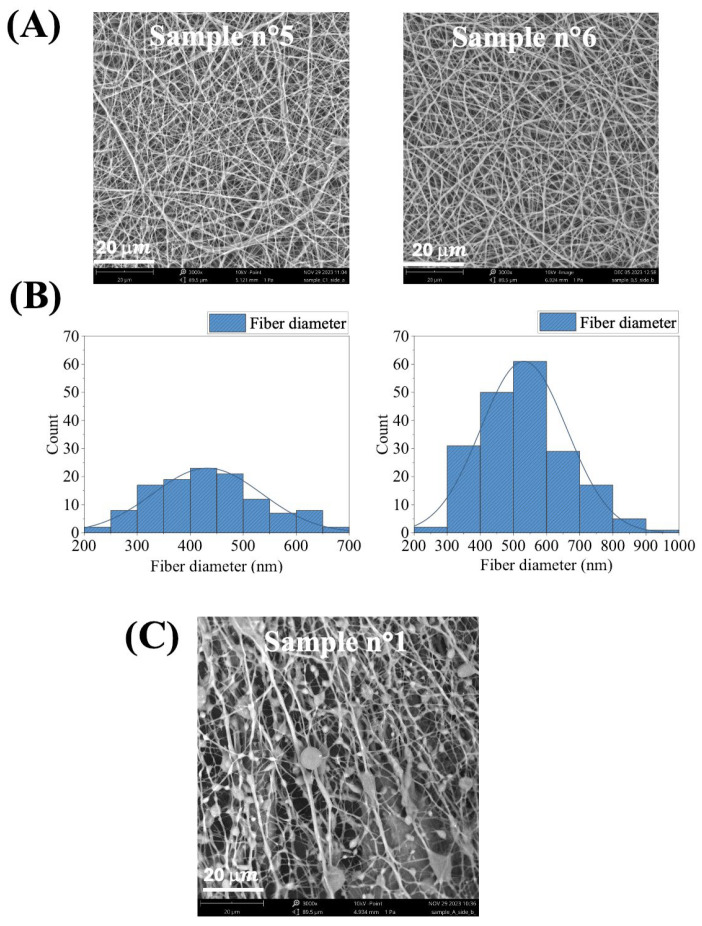
(**A**) SEM images of representative WPU-GSSG_0.5_/PEO electrospun samples. (**B**) Fiber diameter distribution of the WPU-GSSG_0.5_/PEO electrospun membranes calculated from the SEM images using ImageJ (version 1.53e) (*n* = 200). (**C**) SEM images of a representative WPU-GSSG_1.0_/PEO electrospun scaffold.

**Figure 3 ijms-26-11556-f003:**
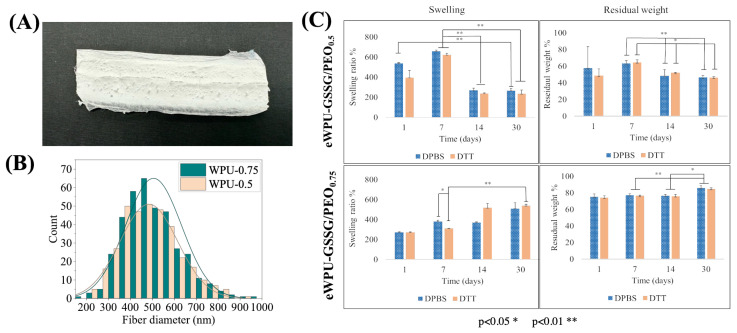
(**A**) Image of the eWPU-GSSG_0.5_/PEO membrane. (**B**) Comparison between electrospun membranes prepared from eWPU-GSSG_0.5_/PEO and eWPU-GSSG_0.75_/PEO. The fiber dimensional distribution was calculated from SEM micrographs using ImageJ (version 1.53e). (**C**) Swellling behavior and percentage of residual weight after hydrolytic degradation of eWPU-GSSGs/PEO electrospun membranes following incubation in DPBS and DPBS containing DTT (* *p* < 0.05 and ** *p* < 0.01).

**Figure 4 ijms-26-11556-f004:**
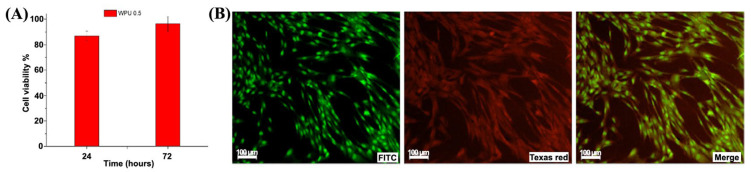
Indirect evaluation of fibroblast viability after 3 days of culture with eWPU-GSSG_0.5_/PEO membranes. (**A**) Quantitative assessment of cell viability using the MTS assay. (**B**) Fluorescence micrographs of fibroblasts.

**Figure 5 ijms-26-11556-f005:**
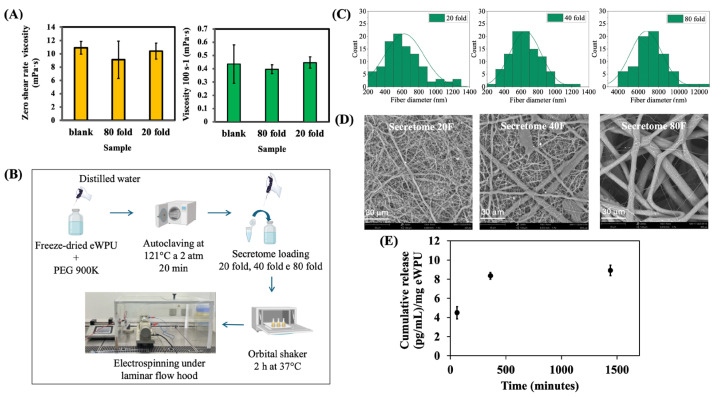
(**A**) Zero-shear-rate viscosity and viscosity of WPU-GSSG_0.5_/PEO dispersions, with or without secretome loading, showing no significant differences. (**B**) Schematic of secretome loading procedure and dispersion preparation prior to electrospinning. (**C**) Fiber diameter distribution and (**D**) representative SEM micrographs of electrospun membranes. (**E**) VEGF-A release from electrospun WPU-GSSG_0.5_/PEO nanofiber membranes loaded with secretome at 20-fold concentration, with an encapsulation efficiency of 93%.

**Table 1 ijms-26-11556-t001:** Molar ratios of diols, isophorone diisocyanate (IPDI) and GSH extender along with the amount of COOH and GSSG incorporated into the WPU derivatives.

WPU	PCL-PEG-PCL/DMPA/IPDI/GSSG Molar Ratio	mg GSSG/gWPU	mg COOH/gWPU
1.00	1/1/3.5/1.5	76.45 ± 3.80	17.96 ± 0.41
0.75	0.75/1.25/3.5/1.5	72.45 ± 5.02	30.10 ± 0.27
0.50	0.5/1.5/3.5/1.5	66.30 ± 4.53	41.87 ± 0.55

**Table 2 ijms-26-11556-t002:** Summary of the electrospinning test batches, polymer concentrations, and the corresponding electrospinning parameters. Electrospinning trials were performed using increasing WPU concentrations ranging from 13% to 29% *w*/*w*, while maintaining the polymeric carrier (PEO 900 kDa) concentrations at 0.5% to 2.5% *w*/*w*. The final row provides a qualitative evaluation of fiber morphology, dimensional uniformity, and the presence of beads or polymer droplets.

Sample Name	WPU-GSSG 0.75 [% *w*/*w*]	PEO [% *w*/*w*]	Q (mL/min)	V (kV)	d (cm)	Fiber Morphology
*1*	12.8	2.12	0.008	20	15	Bead on a string
*2*	16.83	0.58	/	20	15	Non-uniform fiber
*3*	19.5	0.64	/	20	15	Non-uniform fiber
*4*	22.9	0.76	0.008	20	15	Non-uniform fiber
*5*	22.9	0.76	0.02	20	15	Bead on a string
*6*	28.46	0.92	/	20	15	Bead on a string
*7*	22.73	1.52	0.02	28.5	15	Non-uniform fiber
*8*	22.73	1.52	0.02	28.5	20	Non-uniform fiber
*9*	22.5	2.26	0.01	30	30	Bead-free fiber
*10*	22.5	2.26	0.01	22	20	Bead-free fiber
*11*	22.5	2.26	0.008	20	30	Bead-free fiber
*12*	22.5	2.26	0.01	25	25	Bead-free fiber
*13*	22.5	2.26	0.02	25	25	Bead-free fiber
*14*	22.5	2.26	0.01	30	20	Bead-free fiber
*15*	22.5	2.26	0.028	22	25	Bead-free fiber
*16*	22.5	2.26	0.015–0.03	20	25	Bead-free fiber

**Table 3 ijms-26-11556-t003:** WPU-GSSG_0.5_/PEO electrospinning test batches, polymer concentrations, and the corresponding electrospinning parameters. Electrospinning trials were performed using increasing concentrations of WPU 0.5 ranging from 20% to 28% *w*/*w*, while maintaining the polymeric carrier concentration between 2% and 2.5% *w*/*w*. The final row provides a qualitative evaluation of fiber morphology, dimensional uniformity, and the presence of beads or polymer droplets.

Sample Name	WPU-GSSG 0.5 [% *w*/*w*]	PEO [% *w*/*w*]	Q (mL/min)	V (kV)	d (cm)	Fiber Morphology
*1*	27.78	2.78	/	/	/	Non-uniform fiber
*2*	22.56	2.26	/	/	/	Non-uniform fiber
*3*	19.61	1.96	25	25	24	Non-uniform fiber
*4*	20.82	2.08	25	25	20	Bead on a string
*5*	22.5	2.26	20–25	20–25	20–23	Bead-free fiber
*6*	22.5	2.26	22.5	22.5	25	Bead-free fiber
*7*	22.5	2.26	23	23	25	Bead-free fiber
*8*	22.5	2.26	18–20	18–20	20	Bead-free fiber
*9*	22.5	2.26	19	19	15	Bead-free fiber
*10*	22.5	2.26	23	23	20	Bead-free fiber
*11*	22.5	2.26	23	23	20	Bead-free fiber

**Table 4 ijms-26-11556-t004:** WPU-GSSG_1.0_/PEO electrospinning test batches, biopolymer concentrations, and the corresponding electrospinning parameters. Electrospinning trials were performed using a WPU 1.0 concentration of 16% *w*/*w* and a polymeric carrier concentration of approximately 2% *w*/*w*. The final row provides a qualitative evaluation of fiber morphology and the presence of beads.

Sample Name	WPU-GSSG 1 [% *w*/*w*]	PEO [% *w*/*w*]	Q (mL/min)	V (kV)	d (cm)	Fiber Morphology
*1*	16.39	1.64	0.03	27	20	Bead on a string

**Table 5 ijms-26-11556-t005:** Summary of the final electrospinning parameters optimized for WPU-GSSG/PEO membranes.

Sample Name	WPU-GSSG_0.5_ [% *w*/*w*]	PEO [% *w*/*w*]	Q (mL/min)	V (kV)	d (cm)	Fiber Morphology
*eWPU-GSSG_0.5_/PEO*	22.5	2.26	0.02	22.5	25	Bead-free fiber
*eWPU-GSSG_0.75_/PEO*	22.5	2.26	0.028	22	25	Bead-free fiber
*eWPU-GSSG_1.0_* */PEO*	16.39	1.69	0.03	27	20	Bead on a string

**Table 6 ijms-26-11556-t006:** Summary of the optimized electrospinning conditions under a laminar flow hood for three samples prepared with 20-, 40-, and 80-fold secretome concentrations. The final row reports the quantitative fiber diameter.

Sample Name	WPU [% *w*/*w*]	PEO [% *w*/*w*]	Q (mL/min)	V (kV)	d (cm)	Fiber Morphology	Fiber Diameter (µm)
*eWPU-GSSG_0.5_/PEO* *20 fold*	22.5	2.26	0.01	23	10−15	Bead-free fiber	0.658 ± 258.58
*eWPU-GSSG_0.5_/PEO* *40 fold*	22.5	2.26	0.01	25	17	Bead-free fiber	6.41 ± 2.97 × 10^3^
*eWPU-GSSG_0.5_/PEO* *80 fold*	22.5	2.26	0.01	26	20	Bead-free fiber	6.78 ± 1.56 × 10^3^

## Data Availability

The original contributions presented in this study are included in the article. Further inquiries can be directed to the corresponding author.

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
