# Peer review of "Eco-Friendly Fabrication of Secretome-Loaded, Glutathione-Extended Waterborne Polyurethane Nanofibers"

_ijms, 2025, doi:10.3390/ijms262311556_

Round 1

Reviewer 1 Report

Comments and Suggestions for Authors

The authors have successfully introduced the concept of green chemistry (all aqueous phase, no organic solvent) into the process of biomaterial preparation, and are committed to solving the bottleneck problems of poor stability and rapid removal faced by the direct application of secretome. The research design is reasonable, the experimental data are detailed, and the demonstration process is basically clear, which provides a valuable exploration for the development of the next-generation cell-free regenerative therapy platform.

  1. The experimental results showed that VEGF-A was basically completely released within 24 hours (~1500 minutes). This rapid release mode may not meet the demand for sustained and controllable release of bioactive factors in many tissue regeneration applications. The limitations of this sudden release phenomenon should be discussed, and the possible reasons should be analyzed.
  2. Although the rheological data showed that the addition of secretome had little effect on the zero shear viscosity, the SEM results clearly showed that the fiber morphology and diameter distribution had changed significantly. This indicates that there is a complex interaction between the secretome and the spinning solution, and the current manuscript's interpretation of this ("affecting the balance of electrostatic and intermolecular forces") is slightly general.
  3. For tissue engineering scaffolds, mechanical properties are crucial parameters that affect the mechanical sensing and in vivo integration of cells. The characterization of the mechanical properties of the final electrospun membranes is missing in the manuscript.
  4. Part of the discussion can be more refined, highlighting the findings and comparisons most relevant to this study.

Author Response

Question 1: The authors have successfully introduced the concept of green chemistry (all aqueous phase, no organic solvent) into the process of biomaterial preparation, and are committed to solving the bottleneck problems of poor stability and rapid removal faced by the direct application of secretome. The research design is reasonable, the experimental data are detailed, and the demonstration process is basically clear, which provides a valuable exploration for the development of the next-generation cell-free regenerative therapy platform.The experimental results showed that VEGF-A was basically completely released within 24 hours (~1500 minutes). This rapid release mode may not meet the demand for sustained and controllable release of bioactive factors in many tissue regeneration applications. The limitations of this sudden release phenomenon should be discussed, and the possible reasons should be analyzed.

Response 1: We thank the reviewer for this comment. The main objective of this study was to establish a procedure for electrospinning a polyurethane-based biomaterial in an aqueous environment and to demonstrate its suitability for incorporating secretome fractions. In this context, VEGF-A was used as a representative factor to validate both the electrospinning process and the loading capability of the material. The release profile observed in this work is faster than what we previously reported for an HA-based hydrogel, in which a plateau of VEGF release in non-heparinized hydrogels was reached after approximately 150 hours (9000 minutes) (see reference 21 of the revised manuscript version). Compared with the hydrogel system, the faster release observed here is influenced by both the amount and the modality of loading (impregnation versus incorporation during the fabrication process). It is also strongly affected by the high surface-to-volume ratio typical of electrospun fibrous scaffolds, which shortens the diffusional path length relative to bulk 3D hydrogels. We agree that a more sustained and controllable release would be advantageous for many tissue-regeneration applications. Achieving such modulation will likely require the design of composite electrospun scaffolds incorporating glycosaminoglycans (e.g., heparin) or other affinity-based components capable of binding and retaining cytokines. We are currently developing a post-fabrication functionalization strategy that exploits the reversible S-S bonds within the membrane to introduce such bioactive moieties, which we expect will enable a more sustained release profile in future iterations of this material. To satisfy reviewer request we introduced a discussion (see lines 337-349) in the revised manuscript version.

Question 2: Although the rheological data showed that the addition of secretome had little effect on the zero shear viscosity, the SEM results clearly showed that the fiber morphology and diameter distribution had changed significantly. This indicates that there is a complex interaction between the secretome and the spinning solution, and the current manuscript's interpretation of this ("affecting the balance of electrostatic and intermolecular forces") is slightly general.

Response 2: We thank the reviewer for this observation. We expanded the discussion to provide a more detailed hypothesis explaining the impact of secretome loading on spinnability, despite the minimal effect on zero-shear viscosity (see Paragraph 2.3, lines 312-323, of the revised manuscript). The stability of zero-shear viscosity suggests that the cohesive forces within the polymer dispersion are not substantially altered. However, successful electrospinning depends not only on viscosity but also on the balance between cohesive forces and the electrostatic forces acting on the jet. Cytokines and growth factors contain both acidic and basic amino acids, and their incorporation into the spinning solution may influence its overall charge distribution. Under our processing parameters, it is possible that the introduction of these charged biomolecules disrupted the charge balance sufficiently to prevent the establishment of a new stable equilibrium that would support efficient jet elongation. The observed increase in fiber diameter may therefore reflect a reduction in the net charge density of the electrospinning solution, likely due to the polyelectrolytic nature of the polypeptide components in the secretome. A lower charge density would reduce the stretching forces acting on the jet, resulting in thicker fibers even in the absence of viscosity changes.  

Question 3: For tissue engineering scaffolds, mechanical properties are crucial parameters that affect the mechanical sensing and in vivo integration of cells. The characterization of the mechanical properties of the final electrospun membranes is missing in the manuscript.

Response 3: We thank the reviewer for this comment. The aim of the present study was to establish the electrospinning conditions and to verify the suitability of the material for secretome loading and controlled release. For this reason, the mechanical characterization of the final electrospun membranes was not included at this stage.

Question 4: Part of the discussion can be more refined, highlighting the findings and comparisons most relevant to this study.

Response 4: By answering to the reviewer’s questions (rounds 1-3) the results and discussions were further improved, introducing comparisons with other systems.

Reviewer 2 Report

Comments and Suggestions for Authors

The manuscript is "Eco-friendly fabrication of secretome loaded, glutathione extended waterborne polyurethane nanofibers ".

  This article explores a waterborne polyurethane (WPU) via an entirely aqueous electrospinning procedure, providing a solvent-free, green approach to produce nanofibrous matrices suitable for secretome delivery. However, the content and data are not enough to discuss its differences and applications.

  1. All graphs are blurry, making it difficult to distinguish parameter values ​​and scales.
  2. How is optimal balance determined? (See lines 120-122)
  3. Was the viscosity of all polymer solutions used in electrospinning measured? What is the basis for choosing PEO 900 kDa? (See lines 123-125)
  4. The graphic labeling is incorrect on lines 229-245.
  5. 40-fold concentrated secretome can affect electrospinning efficiency, raising the question of whether WPU-GSSG0.5/PEO is unsuitable for drug encapsulation in regenerative medicine.
  6. Can waterborne polyurethane Adding only 150 μL of secretome to a polymer solution via electrospinning and then performing a release test raises the question: could such a small dosage have a significant impact from operational errors? be used to spin smaller diameter nanofibers using finer needles?
  7. What is the mechanical strength of waterborne polyurethane nanofibers? What are their hydrophilicity/hydrophobicity?
  8. Adding only 150 μL of secretome to a polymer solution via electrospinning and then performing a release test raises the question: could such a small dosage have a significant impact from operational errors?

Author Response

Question 1: All graphs are blurry, making it difficult to distinguish parameter values ​​and scales.

Response 1: We thank the reviewer for this suggestion. We have improved the resolution and clarity of all figures to ensure that parameter values and scales are easily distinguishable. In addition, to simplify the figures and enhance readability, we have extracted the tables providing them as separate within the manuscript.

Question 2: How is optimal balance determined? (See lines 120-122)

Response 2: We thank the reviewer for this question: we agree that the expression was unclear. Our intention was simply to indicate that we initiated the electrospinning experiments using the WPU-GSSH with the mean values of diol molar ratios (Table I). Since this information can be confusing and considering that does not contribute to the interpretation of the results, we have removed the sentence.

Question 3: Was the viscosity of all polymer solutions used in electrospinning measured? What is the basis for choosing PEO 900 kDa? (See lines 123-125)

Response 3: We thank the reviewer for this insightful comment. We measured the viscosity of the WPU-GSSG 0.5/PEO dispersions at the concentrations selected for electrospinning, both in the absence and in the presence of secretome. In the revised version of the manuscript, these results are now reported in Figure 5. PEO (900 kDa) is commonly used as a carrier polymer to enhance the spinnability of water-soluble polysaccharides. In our system, the colloidal dispersions of WPU-GSSG derivatives alone did not provide sufficient chain entanglement to enable stable electrospinning, as confirmed by preliminary experiments. The addition of a small amount of high-molecular-weight PEO, present at a concentration ten times lower than that of WPU-GSSG, provides the required entanglements and allows successful and continuous electrospinning. This comment has been added in the revised manuscript (lines 124-129).

Question 4: The graphic labeling is incorrect on lines 229-245.

Response 4: We revised the labelling on the lines evidenced by the reviewer.

Question 5: 40-fold concentrated secretome can affect electrospinning efficiency, raising the question of whether WPU-GSSG0.5/PEO is unsuitable for drug encapsulation in regenerative medicine.

Response 5: We thank the reviewer for this comment. In our experiments, the electrospinning efficiency was partially compromised by increasing the concentration of the loaded secretome (See response to reviewer 1). In fact, we did observe a marked increase in fiber diameter at higher secretome concentrations, most likely due to alterations in the electrical balance during the electrospinning process. This modification of fiber morphology does not inherently limit the suitability of the WPU-GSSG0.5/PEO system for drug encapsulation. Rather, the larger fiber diameter may influence the surface area of the membrane and therefore the release kinetics of the encapsulated factors. In general, each specific therapeutic agent should be evaluated individually, as its physicochemical properties may differently influence spinnability and final fiber morphology. 

Question 6: Can waterborne polyurethane Adding only 150 μL of secretome to a polymer solution via electrospinning and then performing a release test raises the question: could such a small dosage have a significant impact from operational errors? be used to spin smaller diameter nanofibers using finer needles?

Response 6: We thank the reviewer for raising this point. We understand that the concern may arise from an imprecise description in paragraph 3.6. In fact, the electrospinning solution was prepared using 600 mg of WPU-GSSG0.5 and 60 mg of PEG (900 kDa), dispersed in water to obtain final concentrations of 22.5% and 2.26%, respectively, corresponding to a total volume of 2.6 mL. The addition of 150 μL of secretome was therefore made to a relatively large and well-defined volume, ensuring an accurate handling, reproducible electrospinning, and reliable collection. We apologize for the lack of clarity in the original description. The experimental section has now been revised to provide a precise explanation of the volumes and concentrations used.   

Question 7: What is the mechanical strength of waterborne polyurethane nanofibers? What are their hydrophilicity/hydrophobicity?

Response 7: The aim of the present study was to optimize the electrospinning conditions and to assess the suitability of the waterborne polyurethane (WPU) material for secretome loading and controlled release. For this reason, the mechanical characterization of the final electrospun membranes was not included at this stage. However, we are currently developing a secretome-loaded WPU electrospun membrane for preclinical evaluation in wound-healing applications, and in that ongoing project we will carry out a comprehensive mechanical assessment of the biomaterial. Regarding hydrophilicity, the wettability of the WPU-GSSG materials was already evaluated in our previous manuscript. After 48 h, WPU-GSSG1, WPU-GSSG0.75 and WPU-GSSG0.5 exhibited water contact angles of 71°, 65° and 55°, respectively. These results indicate progressively increasing hydrophilicity, which correlates with the higher content of DMPA from WPU-GSSG1 to WPU-GSSG0.5. The increased DMPA results in a higher concentration of carboxyl groups, thereby enhancing the hydrophilicity of the polymers (see Reference 28).

Question 8: Adding only 150 μL of secretome to a polymer solution via electrospinning and then performing a release test raises the question: could such a small dosage have a significant impact from operational errors?

Response: as explained in the response 6 this concern probably arises from an imprecise description in paragraph 3.6. In fact, the electrospinning solution was prepared using 600 mg of WPU-GSSG0.5 and 60 mg of PEG (900 kDa), dispersed in water to obtain final concentrations of 22.5% and 2.26%, respectively, corresponding to a total volume of 2.6 mL. The addition of 150 μL of secretome was therefore made to a relatively large and well-defined volume, thus avoiding possible operational errors.

Reviewer 3 Report

Comments and Suggestions for Authors

The manuscript “Eco-friendly fabrication of secretome loaded, glutathione extended waterborne polyurethane nanofibers” describes the achieving of nanofibrous matrices by a green approach involving the electrospinning of aqueous solutions of waterborne polyurethanes. Taking into account that the synthesis and characterization of the polymers were published in a previous work, in this manuscript the authors mainly focused on the optimization of the electrospinning process, testing of the electrospinned membrane cytocompatibility and the loading of secretome in various proportions in the electrospinned fibers.

The study is systematic, the results are relevant and clearly presented, so I have only some minor observations:

  • The tables included in the figures (Fig.1B, Fig. 2A, D,..) are very small and difficult to follow, so perhaps the authors could find another possibility to represent them (even as tables in the main text).
  • Regarding Fig.3, there are some discrepancies between the images, the figure caption and the text. Please revise.
  • 5 is very small and unclear. I am not sure, but I think also the value of fiber diameter for the second sample (40 fold) is wrong.
  • In the Materials section are included toluene, diethyl ether, isopropanol. Where these solvents are used in the experimental protocols, as for example, toluene is highly toxic?

Author Response

The study is systematic, the results are relevant and clearly presented, so I have only some minor observations:

Question 1: The tables included in the figures (Fig.1B, Fig. 2A, D,..) are very small and difficult to follow, so perhaps the authors could find another possibility to represent them (even as tables in the main text).

Response 1: We have improved the resolution and clarity of all figures to ensure that parameter values and scales are easily distinguishable. In addition, to simplify the figures and enhance readability, we have extracted the numerical data and provided them as separate tables within the manuscript.

Question 2: Regarding Fig.3, there are some discrepancies between the images, the figure caption and the text. Please revise.

Response 2: As suggested, we revised the Figure, clearing discrepancies.

Question 3: 5 is very small and unclear. I am not sure, but I think also the value of fiber diameter for the second sample (40 fold) is wrong.

Response 3: We revised Figure 5 making it more clear and improving the resolution.

Question 4: In the Materials section are included toluene, diethyl ether, isopropanol. Where these solvents are used in the experimental protocols, as for example, toluene is highly toxic?

Response 4: These solvents are routinely used in established procedures for PEG–PCL–PEG synthesis. In particular, toluene is employed only in very small amounts to enable the azeotropic distillation of PEG, ensuring complete removal of water prior to initiating the reaction. The toluene is fully removed during this distillation step. Afterwards, several processing and purification steps are performed before obtaining the final PCL-PEG-PCL product, ensuring that no residual toxic solvents remain in the material.

Round 2

Reviewer 2 Report

Comments and Suggestions for Authors

The author has provided explanations and improvements regarding the comments.